# Biogenic Approaches to Metal Nanoparticle Synthesis and Their Application in Biotechnology

**DOI:** 10.3390/plants15020183

**Published:** 2026-01-07

**Authors:** Yulia Yugay, Yury Shkryl

**Affiliations:** Federal Scientific Center of the East Asia Terrestrial Biodiversity of the Far East Branch of Russian Academy of Sciences, 159 Stoletija Str., Vladivostok 690022, Russia

**Keywords:** biogenic nanoparticles, plant cell and tissue cultures, bioengineering of reductive capacity, secondary metabolite elicitation, plant–nanoparticle interactions

## Abstract

Metal and metal oxide nanoparticles (NPs) synthesized through biologically mediated reduction of metal ions using biomolecules derived from microorganisms, algae, or plants are attracting growing attention in plant biotechnology due to their multifunctional properties and environmental advantages compared with conventional physicochemical synthesis. This review provides a comprehensive analysis of biological approaches for NP production using bacteria, fungi, algae, cyanobacteria, whole plants, and in vitro plant cell cultures. The main biosynthetic mechanisms, types of reducing and capping metabolites, metal specificity, and typical NP characteristics are described for each system, with emphasis on their relative productivity, scalability, reproducibility, and biosafety. Special consideration is given to plant cell and tissue cultures as highly promising platforms that combine the metabolite diversity of whole plants with precise control over growth conditions and NP parameters. Recent advances highlight the significance of bioengineering of reductive capacity as a novel strategy to enhance the efficiency and controllability of NP biosynthesis. Since NP formation is driven by key biomolecules, targeted modification of biosynthetic pathways through metabolic and genetic engineering can substantially increase NP yield and allow fine-tuning of their structural and functional properties. The applications of biogenic NPs in plant biotechnology are systematically evaluated, including their use as environmentally safe disinfectants for explants and seed sterilization, modulators of callus induction and morphogenesis, and abiotic elicitors that enhance the accumulation of economically valuable secondary metabolites. Remaining challenges, such as variability in NP characteristics, limited scalability, and insufficient data on phytotoxicity and environmental safety, are discussed to outline future research priorities. The synthesis–function relationships highlighted here provide a foundation for developing sustainable NP-based technologies in modern agriculture.

## 1. Introduction

The rapid growth of global agriculture and biotechnology has intensified the demand for innovative tools that enhance plant productivity, disease resistance, and metabolite synthesis while minimizing environmental impact. Conventional chemical treatments, antibiotics, and synthetic elicitors used in plant tissue culture and crop protection often exhibit limited efficiency, phytotoxicity, and ecological risks, including the emergence of microbial resistance and accumulation of toxic residues [1,2,3]. To overcome these challenges, nanotechnology has emerged as a transformative field capable of manipulating matter at the nanoscale to create materials with unique physicochemical and biological properties [4,5]. Among various nanostructured materials, metal and metal oxide NPs have attracted particular attention due to their high surface area, catalytic activity, and tunable reactivity, enabling a broad spectrum of applications in medicine, environmental science, and agriculture [3,6,7].

However, conventional physical and chemical NP synthesis methods frequently involve toxic reagents, high energy consumption, and complex waste management, contradicting the goals of sustainable biotechnology [5]. Consequently, there is a pressing need for eco-friendly, cost-effective, and biocompatible alternatives that can produce NPs under mild, aqueous, and scalable conditions. Biogenic or “green” synthesis addresses this challenge by utilizing living organisms or their metabolites as natural nanofactories capable of reducing metal ions and stabilizing the resulting NPs through biomolecules such as enzymes, proteins, polysaccharides, and polyphenols [8]. These biosynthetic systems, which include bacteria, fungi, algae, cyanobacteria, higher plants, and in vitro plant cell cultures, embody the principles of green chemistry and demonstrate remarkable catalytic versatility and environmental compatibility.

Each biological system contributes distinct biochemical pathways and reducing agents, yielding NPs with diverse morphologies, compositions, and functions. Bacteria and fungi have long served as efficient producers of silver, gold, and copper NPs through the action of NADPH-dependent reductases and extracellular proteins [9,10]. Algae and cyanobacteria offer additional sustainability advantages, using photosynthetic pigments, polysaccharides, and phenolics to generate metal and metal oxide NPs [11,12,13]. Higher plants, rich in secondary metabolites, have proven capable of synthesizing NPs of nearly all major metals, while in vitro plant cell and tissue cultures provide an advanced platform combining high biosynthetic potential with strict control over environmental and biochemical parameters [14,15,16]. These systems collectively enable reproducible and sustainable NP production without reliance on hazardous chemicals.

Beyond synthesis, biologically derived NPs have shown exceptional promise as functional agents in plant biotechnology. In tissue culture, silver, zinc oxide, and titanium dioxide NPs act as broad-spectrum disinfectants and antibiotic substitutes, reducing contamination while avoiding phytotoxic effects [17,18]. When introduced into culture media, Ag NPs can enhance callus formation, organogenesis, and somatic embryogenesis by modulating ethylene biosynthesis, redox status, and antioxidant enzyme activity [19,20,21]. Moreover, metal and metal oxide NPs function as abiotic elicitors, triggering signaling cascades such as MAPK and Ca^2+^-dependent pathways and thereby stimulating the accumulation of secondary metabolites, including phenolics, flavonoids, terpenoids, and alkaloids [22,23]. These elicitation effects not only enhance the productivity of valuable phytochemicals in callus, suspension, and hairy root cultures but also provide new strategies for developing stress-tolerant and high-value plant systems.

Despite this progress, several challenges hinder the broader application of biogenic NPs in agriculture and biotechnology. These include relatively low and variable yields, difficulties in controlling NP size and surface properties, and incomplete understanding of the molecular mechanisms underlying NP nucleation, growth, and plant–NP interactions [24,25]. Addressing these challenges requires an integrated approach combining omics-based profiling, metabolic and genetic engineering of producer organisms, and standardization of synthesis and characterization protocols. The development of reliable toxicological assessment frameworks and environmentally responsible deployment strategies is equally critical to ensure biosafety and regulatory acceptance [26,27]. This review provides a comprehensive analysis of biological approaches to the synthesis of metal and metal oxide NPs and their expanding applications in plant biotechnology.

## 2. Biological Methods for Metal NP Synthesis

Despite the wide range of physical and chemical approaches available for NP synthesis, there is a growing demand for alternative strategies that are more sustainable and adaptable. In this context, increasing attention has been directed toward biotechnological methods for metal NP production, in which living organisms, biological systems, or their derivatives, including crude fractions and purified biomolecules, act as catalysts for the synthesis process. In addition to cost-effectiveness and environmental compatibility, biological approaches offer several advantages over conventional methods, such as high catalytic efficiency in aqueous media under ambient temperature and pressure, as well as considerable process flexibility that enables implementation under diverse conditions and across different scales (Table 1) [28].

At the mechanistic level, biological synthesis of metal NPs relies on the ability of biological systems to mediate the transformation of metal ions from soluble precursors into nanoscale solid phases, followed by nucleation and controlled particle growth. For noble and some transition metals (e.g., Ag, Au, Pd, and Pt), this process predominantly involves redox-driven reduction of metal ions into zero-valent or low-valent states, whereas for metal oxides (e.g., ZnO, Fe_2_O_3_, and TiO_2_), it often proceeds through non-redox pathways such as biomolecule-assisted hydrolysis, precipitation, and controlled crystallization. This multistep process is governed by a broad spectrum of biomolecules that function as reducing agents, stabilizers, and structure-directing components. These include enzymes and low-molecular-weight metabolites, as well as macromolecules such as proteins and polysaccharides [29,30]. Enzymatic components typically enable specific and kinetically efficient electron transfer, often coupled to cellular redox metabolism [31], whereas non-enzymatic metabolites mediate metal ion reduction through reactive functional groups such as hydroxyl, carbonyl, thiol, or amine moieties, facilitating both metal chelation and electron donation [32]. Macromolecules, particularly proteins and polysaccharides, predominantly contribute to NP stabilization by forming capping layers that inhibit aggregation and modulate particle size, morphology, and surface charge [33]. Despite the functional diversity of these biomolecular participants, the overall mechanism of biological NP formation generally follows a conserved sequence of metal ion binding, bioreduction, nucleation, and growth, with system-specific differences arising mainly from the qualitative and quantitative composition of the involved biomolecules rather than from fundamentally distinct chemical pathways.

## 3. Bacteria-Mediated NP Synthesis

Among the biological entities used for metal NP synthesis, microorganisms have been the most extensively studied. Bacteria exhibit remarkable biochemical diversity; many species can be isolated from the environment, and their cultivation conditions can be optimized, making them promising agents for NP synthesis (Table 2). Additionally, bacterial biosynthetic parameters can be controlled through culture conditions such as medium composition, temperature, and aeration. By modifying these factors, it is possible to obtain NPs with desired sizes and morphologies. The biochemical mechanisms underlying bacterial NP synthesis have been partially identified and are currently being actively studied for a more comprehensive understanding of the process. These mechanisms include various modifications of precursor solubility, bioadsorption, bioaccumulation, metal precipitation, and cellular transport systems [25]. These processes collectively contribute to bacterial resistance to inorganic compounds through various enzymatic cascades. A primary group of enzymes involved in these processes includes reductases, such as oxidoreductases, nitrate reductases, and sulfite reductases [9]. A notable example of microbial reducers is magnetotactic bacteria, which form magnetosomes—specialized cellular organelles containing iron oxide and sulfide nanocrystals encapsulated in lipid membranes [34]. Due to their unique properties, magnetosomes have applications in targeted therapy [34], molecular imaging [35], biosensing [36], and other fields.

Overall, using bacteria to synthesize biological NPs represents a promising approach due to their enormous diversity. However, limitations related to the toxicity of metal ions to cells, the pathogenicity of some strains, and the difficulty of obtaining monodisperse particles highlight the need to develop alternative biological systems with higher productivity and improved safety. These advantages are particularly evident in fungi, which have recently become a more scalable and industrially adaptable platform for NP production.

## 4. Fungi-Mediated NP Synthesis

The synthesis of NPs using fungi presents several advantages over bacterial synthesis, including ease of scaling, economic feasibility, and environmental sustainability [54]. Table 3 provides examples of research focused on NP synthesis using fungi. Fungi contain enzymes that can initiate NP formation both in the cytoplasm and within the cell wall, allowing for both intracellular and extracellular synthesis [55]. Metal ions are adsorbed onto the cell surface due to charge differences between precursor ions and cell wall components. Additionally, fungi secrete large amounts of proteins into the culture medium, enhancing NP synthesis efficiency. Intracellularly synthesized NPs are often smaller than extracellularly formed ones, with intracellular accumulation potentially useful for bioremediation of contaminants [56]. Conversely, extracellular synthesis results in NPs solutions free of unwanted cellular impurities and allows for repeated use of fungal cultures in the synthesis process [57].

A well-characterized reducing biomolecule from fungi is NADPH-dependent nitrate reductase, identified in *Fusarium oxysporum*, which induces the rapid formation of stable, 10–25 nm monodisperse Ag NPs [41]. Another example is an NADPH-dependent reductase isoform found in *Cladosporium cladosporioides*, which, in combination with phenolic compounds, facilitates the formation of quasi-spherical Au NPs ~72 nm [10]. Fungi can also synthesize oxide and other metal NPs, such as copper, zinc oxide, iron, manganese oxides, and cadmium sulfide. *Aspergillus terreus*, for instance, is capable of extracellular synthesis of Ag NPs [58], which have applications in heavy metal removal and biomolecule immobilization [59,60]. However, concerns regarding pathogenic fungi like *F. oxysporum* and *Trichoderma* species highlight the need for safer alternatives [41,60,61,62,63].

**Table 3 plants-15-00183-t003:** Obtaining metal and metal oxide NPs using fungi.

Organism	Metal	Size (nm), Shape	References
*Trichoderma harzianum*	Ag	1–30, spherical	[60,61,62]
*Aspergillus niger*	Ag	10–100, spherical	[64]
*Penicillium duclauxii*	Ag	3–32, spherical	[65]
*Guignardia mangiferae*	Ag	5–30, spherical	[66]
*Aspergillus versicolor*	Ag	5–39, spherical	[67]
*Pleurotus ostreatus*	Au	10–30, spherical	[68]
*Trichoderma harzianum*	Au	32–44, spherical	[63]
*Yarrowia lipolytica NCIM 3589*	Au	9–27, polymorphic	[69]
*Aspergillus niger*	ZnO	53–69, spherical	[70]
*Saccharomyces cerevisiae*	ZnO	50–70, hexagonal	[71]
*Rhodotorula mucilaginosa*	CuO	10–50, spherical	[72]
*Candida albicans*	Fe_2_O_3_	80, spherical	[73]
*Saccharomyces cerevisiae*	MnO_2_	15–70, hexagonal and spherical	[74]

Although fungi usually provide higher yields and better scalability than bacteria, concerns remain regarding biosafety and purification when working with pathogenic species. This has driven increasing interest toward algae and cyanobacteria, which combine rapid growth, an abundance of reducing metabolites, and minimal biosafety concerns while enabling the synthesis of diverse metal and metal oxide NPs.

## 5. Algae- and Cyanobacteria-Mediated NP Synthesis

Algae and cyanobacteria represent promising biological platforms for the “green” synthesis of metal NPs, offering an eco-friendly and sustainable alternative to traditional methods. Algae, a member of the kingdom *Protista*, include a wide variety of autotrophic and photosynthetic organisms, ranging from unicellular microalgae to complex multicellular macroalgae. Their ability to mediate NP formation is due to the abundance of bioactive compounds, such as polysaccharides (alginate, carrageenan, laminaran, and fucoidan), proteins, pigments, and phenolic molecules, which act as natural reducing agents and stabilizers [11,75,76,77]. Functional groups of these biomolecules, such as hydroxyl, amine, carboxyl, phosphate, and sulfate, promote both the reduction of metal ions and the binding of NPs, enhancing colloidal stability and biocompatibility.

Macro- and microalgae differ in their biosynthetic potential. For macroalgae, aqueous extracts or isolated polysaccharide fractions are typically used, as the composition of bioactive compounds varies depending on the species and environmental conditions. Microalgae, including diatoms and green algae, have demonstrated significant efficiency in the production of metal and composite NPs (Table 4). For example, diatoms (*Nannochloris atomus*, *Diadesmis gallica*) have been used to synthesize Au NPs and Au/Si nanobiocomposites using their silicon frustules as natural templates [78]. Similarly, *Chlorella vulgaris* has been shown to produce Ag and Pd NPs with potent antimicrobial and catalytic activities [79,80]. Cyanobacteria are photosynthetic prokaryotes that combine rapid growth, metabolic diversity, and the ability to fix atmospheric nitrogen. Their nitrogen-fixing enzyme, nitrogenase, is thought to mediate metal ion reduction through its electron-transfer capacity, thereby facilitating NP nucleation and controlling their growth [12]. In addition to intact cultures, cell-free cyanobacterial extracts can efficiently synthesize NPs such as Ag, Au, and ZnO with tunable optical and antimicrobial properties [13,81]. Cyanobacteria such as *Anabaena*, *Spirulina platensis*, and *Nostoc* have been used to biosynthesize metal and metal oxide NPs. These systems exhibit high tolerance for metal ions and can simultaneously bioreduce and stabilize them via extracellular polysaccharides and pigment-associated proteins such as phycocyanin. The resulting NPs often exhibit enhanced antioxidant, antimicrobial, and photocatalytic activities, expanding their potential applications in biomedicine, environmental remediation, and renewable energy.

**Table 4 plants-15-00183-t004:** Production of metal and metal oxide NPs using algae and cyanobacteria.

Organism	Metal	Size (nm), Shape	References
*Spyridia fusiformis*	Ag	5–50, spherical	[82]
*Enteromorpha compressa*	Ag	4–24, spherical	[83]
*Padina pavonia*	Ag	40–80, polymorphic	[84]
*Botryococcus braunii*	Ag, CuO, Cu_2_O	10–40, spherical	[85]
*Cystoseira baccata*	Au	8–40, spherical	[86]
*Stephanopyxis turris*	Au	10–30, spherical	[87]
*Galaxaura elongata*	Au	3–77, polymorphic	[88]
*Chlorella vulgaris*	Pd	5–20, spherical	[79]
*Sargassum ilicifolium*	Al_2_O_3_	10–30, spherical	[89]
*Sargassum wightii*	ZrO_2_	5–20, spherical	[76]

Despite their ecological friendliness and biosynthetic potential, the overall yield of NPs synthesized by these organisms remains relatively low due to variability in metabolic activity, growth phase, and biochemical composition under different culture conditions [24]. This biological variability results in poor reproducibility of NP size, morphology, and surface characteristics. Extracting and purifying biologically active compounds such as polysaccharides, proteins, and pigments is often time-consuming and technically challenging, and the presence of impurities can complicate the isolation process, leading to the formation of polydisperse or unstable NPs. Furthermore, the accumulation of heavy metals in algal biomass creates environmental and production problems related to waste disposal and potential metal release.

## 6. Plant-Mediated NP Synthesis

The potential of plants for NP biosynthesis was first demonstrated by Gardea-Torresdey et al. in 1999 [90], when they described the formation of gold NPs in the biomass of alfalfa (*Medicago sativa*) [91], suggesting a biochemical mechanism. Later, the same researcher showed that alfalfa plants grown in media enriched with gold chloride and silver nitrate were capable of forming corresponding NPs [92]. In both cases, NPs ranging from 2 to 20 nm accumulated within plant stems, particularly along the vascular system. Currently, NP biosynthesis has been studied in a wide range of plant species across different taxa, including *Meliaceae*, *Lamiaceae*, *Lauraceae*, *Asteraceae*, and others (Table 5) [93,94,95,96]. Interestingly, the level of NP accumulation in plant tissues is often significantly higher than in microbial cells, likely due to the plants’ greater tolerance to metal ions. It is suggested that metal accumulation within plants may even provide protection against herbivores and insects. The process of obtaining metal NPs through plants that absorb metals from soil, groundwater, and sediments is known as phytoextraction [97]. Another method for plant-based NP synthesis involves using plant extracts. Various plant extracts, including those from hibiscus leaves (*Hibiscus rosa-sinensis*) [98], black tea [99], Indian gooseberry (*Emblica officinalis*) [100], coffee, and green tea [101], have been successfully used to synthesize gold, silver, nickel, cadmium, lead, chromium, copper, silicon, magnesium, titanium, and zinc NPs with various biological and physicochemical properties (Table 5).

**Table 5 plants-15-00183-t005:** Production of metal and metal oxide NPs using plants.

Organism	Metal	Size (nm), Shape	References
*Piper nigrum*	Ag	9–30, crystalline	[102]
*Phyllanthus emblica*	Ag	20–93, spherical	[103]
*Abelmoschus esculentus*	Au	45–75, spherical	[104]
*Syzygium aromaticum*	Cu	15, spherical	[105]
*Parthenium hysterophorus*	ZnO	28–84, spherical and hexagonal	[96]
*Olea europaea*	ZnO	41–124, crystalline	[106]
*Lycopersicon esculentum*	ZnO	66–133, crystalline	[107]
*Citrus limon*	ZnO, TiO_2_	20–200, polymorphic	[108]
*Phyllanthus amarus*	MnO_2_	40–50, rod-like nano-architectures	[109]
*Matricaria chamomilla*	MgO, MnO_2_	9–112, spherical	[110]
*Rosmarinus officinalis*	MgO	20–68, star-shaped	[111]
*Withania coagulans*	Fe_2_O_3_	16, nanorods	[112]

Crude plant extracts contain diverse bioactive molecules such as proteins, polysaccharides, nucleic acids, and secondary metabolites, with variations between species [113]. Studies on the biochemical mechanisms of NP formation have highlighted the importance of phytochemicals, which play a role in nucleation initiation, growth, and NP stabilization [114]. Each fraction of biomolecules contributes differently to NP biosynthesis. For example, proteins from *Capsicum annuum* leaf extract reduced silver ions to form Ag NPs [115]. Similarly, proteins from *Deinococcus radiodurans* were used to synthesize Au, Ag, and Au/Ag NPs [116].

Plant polyphenols also exhibit strong metal-reducing properties. Flavonoids from *Artemisia tilesii* and *Artemisia annua* extracts demonstrated high reducing activity toward Ag NPs [117]. Moreover, plant polysaccharides are actively used for Ag NP synthesis [118], as well as for bimetallic NPs [119]. While nucleic acids are generally not considered reducers, studies have shown that synthetic oligonucleotides and polynucleotides can reduce metal ions [120]. The shape and size of NPs during biosynthesis can be controlled by adjusting metal and extract concentrations, as well as other reaction conditions [121,122].

While the use of intact plants or plant extracts enables rapid NP formation and high metal tolerance, the influence of developmental stage and environmental factors may limit reproducibility. To overcome this limitation, the focus has shifted toward in vitro plant cell cultures, which offer precisely controlled growth conditions, standardized metabolite profiles, and high reproducibility for NP synthesis.

## 7. Plant Cell Culture-Mediated NP Synthesis

To date, over one hundred plant species have been studied for their ability to biosynthesize metal NPs using extracts obtained from various organs, such as leaves, roots, and seeds. However, the use of in vitro plant cell cultures, particularly callus and suspension cultures, for NP synthesis has only recently gained attention. Callus cultures consist of dedifferentiated, rapidly growing plant cells maintained under sterile conditions on nutrient media with controlled concentrations of plant growth regulators. They represent a reproducible and controllable model system for NP synthesis compared to whole plants, as environmental factors such as light, temperature, and humidity are easily standardized.

The potential of callus cultures for NP production was first demonstrated by Mude et al. [14], who synthesized Ag NPs (60–80 nm) from *Carica papaya* callus extracts. Since then, numerous studies have confirmed the capability of callus cultures to produce metallic and metal oxide NPs, including silver, gold, and zinc oxide, *Linum usitatissimum* [15], *Lithospermum erythrorhizon* [16], and *Viola canescens* [123] (Table 6). The biosynthetic efficiency of callus extracts is attributed to their high content of secondary metabolites such as flavonoids, terpenoids, and phenolic compounds, which act as reducing and capping agents during NP formation. For example, thidiazuron-induced callus extracts of *L. usitatissimum* produced smaller and more uniformly distributed Ag NPs (19–24 nm) compared to whole-plant extracts, due to higher levels of polyphenols and flavonoids [15].

Plant cell cultures possess unique advantages, including high reproducibility, ease of scale-up, and the ability to manipulate biosynthetic pathways through controlled elicitation and genetic modification. Environmental and hormonal parameters can be fine-tuned to optimize the synthesis rate, shape, and size of NPs. Furthermore, elicitation using NPs themselves, such as biogenic ZnO or CuO NPs, has been shown to enhance the production of valuable phytochemicals and secondary metabolites in callus cultures of *Delonix elata* [131] and *Gymnema sylvestre* [132]. The controlled in vitro environment also allows for the development of transgenic cell lines with enhanced metabolite productivity and metal tolerance, providing a sustainable and legally less restricted alternative to field-grown transgenic plants.

The potential applications of NPs synthesized via callus cultures in plant biotechnology and agriculture are extensive. Biogenic NPs can function as nanofertilizers, improving nutrient efficiency and stimulating growth, or as nanopesticides, offering environmentally friendly pest and pathogen control through antimicrobial and antioxidative mechanisms. In addition, their use in phytoremediation, for example, Fe_3_O_4_ [26,133] and ZnO [26,27] NPs synthesized from callus cultures, can enhance heavy metal adsorption and pollutant degradation in contaminated soils and water systems. Callus-derived NPs are also promising as nanocarriers for controlled delivery of agrochemicals, plant hormones, and even genetic material, facilitating precision agriculture and targeted crop improvement. By integrating callus-based nanobiosynthesis with metabolic engineering and omics-driven optimization, researchers can create renewable bioplatforms for producing functional nanomaterials that support plant health, improve resilience, and contribute to sustainable agricultural systems.

Altogether, the evolution from microbial systems to whole plants and plant cell cultures reflects a progressive improvement in control, reproducibility, and biosynthetic efficiency (Figure 1). Biological synthesis strategies differ not only in productivity and scalability but also in the morphological diversity of the NPs produced. Depending on the organism and its metabolites, biogenic routes can yield NPs of various shapes: spherical, cubic, hexagonal, rod-like, or irregular, as well as a wide size range typically spanning from 5 to 200 nm. These structural variations are determined by the nature and concentration of reducing and capping biomolecules such as enzymes, proteins, polysaccharides, and polyphenols, as well as by environmental factors including pH, temperature, and metal ion concentration. In general, bacteria and fungi enable rapid and cost-effective synthesis but often produce polydisperse particles, whereas algae and cyanobacteria provide safer and more environmentally sustainable platforms with broad enzymatic diversity and tunable particle morphologies. Higher plants typically exhibit the greatest metal accumulation capacity and can generate a wide spectrum of NP sizes and shapes due to their complex metabolite composition, while in vitro plant cell cultures offer the highest degree of control and reproducibility, allowing targeted modulation of NP characteristics through culture conditions and elicitation. Understanding these differences enables informed selection of the most appropriate biosynthetic platform for specific applications in biotechnology and agriculture.

## 8. Bioengineering of Reductive Capacity

The formation of biogenic metal NPs is primarily governed by the intrinsic biochemical machinery of the producing organism. Fundamental biomolecules such as proteins, polysaccharides, and low-molecular-weight secondary metabolites serve as reducing and stabilizing agents, mediating the nucleation and growth of NPs under mild, aqueous conditions [16,123]. Therefore, modulation of their biosynthetic pathways through metabolic and genetic engineering represents a promising strategy for enhancing the reductive capacity of biological systems, thereby improving NP yield or altering their physicochemical properties.

The biosynthesis of proteins and polysaccharides, in particular, plays a crucial role in the redox balance of cells and the stabilization of nascent NPs. Enzymes such as reductases, oxidases, and dehydrogenases provide electrons for metal ion reduction, while extracellular polysaccharides and glycoproteins serve as capping agents that prevent aggregation and confer biocompatibility [9,16]. Thus, targeted upregulation of genes encoding such biomolecules can enhance NP productivity. A notable example was demonstrated by overexpression of the silicatein gene, which is responsible for biosilicification in marine sponges. When expressed in *Nicotiana tabacum* callus cultures, silicatein significantly increased the production of Ag NPs compared to non-transformed controls [129]. This result suggests that enzymatic pathways involved in biomineralization can be effectively repurposed to improve metal NP biosynthesis in plant systems.

Activation of secondary metabolism also exerts a profound impact on the reductive potential of plant cell and tissue cultures. Secondary metabolites such as phenolics, flavonoids, terpenoids, and alkaloids are known to possess strong reducing properties due to their multiple hydroxyl and carbonyl groups, which facilitate metal ion reduction [22,114]. In this context, *rolC*-transgenic callus and hairy root cultures of *Panax ginseng* accumulated significantly higher levels of ginsenosides compared to non-transformed controls. These transgenic cultures exhibited an enhanced reduction potential, resulting in increased biosynthetic activity and a greater yield of Ag NPs [134]. The hairy root extract possessed the maximal reductive capacity, and the biosynthesized Ag NPs displayed strong antifungal activity against several *Triticum aestivum* pathogens, confirming the functional advantage of metabolically engineered systems [133].

A similar relationship between secondary metabolism and NP productivity was observed in *Aristolochia manshuriensis* hairy roots. These transgenic lines, characterized by elevated levels of phenanthrene derivatives, produced substantially higher amounts of Ag NPs with pronounced antibacterial and cytotoxic activities [130]. The increased reductive potential of the extracts was attributed to the enrichment of aromatic and phenolic compounds capable of serving as efficient electron donors during NP synthesis. Such findings highlight the critical role of metabolic specialization in determining the efficiency and functional properties of biogenic NPs.

Collectively, these examples demonstrate that the bioengineering of reductive capacity through genetic and metabolic manipulation offers a powerful approach to improving the biosynthesis of metal NPs in plant systems. By enhancing the accumulation of reducing metabolites and catalytic proteins, it is possible to achieve higher productivity, greater monodispersity, and tailored surface characteristics of NPs. Integration of omics-based analyses, gene editing, and metabolic modeling will further enable the rational design of high-performing nanobiosynthetic platforms for sustainable nanotechnology applications in plant biotechnology and agriculture.

## 9. Application of NPs in Plant Tissue Culture

### Explant Disinfection

Bacterial and fungal contamination remains a major problem in the establishment of in vitro plant cultures. Plant material, as well as laboratory glassware, instruments, and culture media, serve as potential sources of microbial infection [135]. Vegetative and reproductive plant organs collected in the field or greenhouse must undergo surface sterilization before being introduced into culture (Figure 2). However, traditional surface disinfection procedures are often insufficient, leading to microbial contamination of biological material and, ultimately, to the loss of the culture [1]. A wide range of chemical disinfectants is used to sterilize explants, including bromine water, sodium or calcium hypochlorite, ethanol, hydrogen peroxide, mercuric chloride, silver nitrate, antibiotics, and fungicides. The choice of sterilizing agent, as well as its concentration and exposure time, significantly affects the viability of explants and subsequent morphogenesis. High concentrations or prolonged exposure often cause oxidative stress and tissue necrosis, especially in sensitive meristematic zones. Moreover, the elimination of endophytic microorganisms remains challenging, often requiring prolonged inclusion of antibiotics in the culture medium. However, antibiotics are known to have phytotoxic effects on plant cells both in vitro and in vivo and can impair metabolic processes or the ability to regenerate [136,137]. Moreover, prolonged exposure to antibiotics promotes the development of inherited bacterial resistance [2,3].

In recent years, the application of metal and metal oxide NPs has emerged as an effective and less phytotoxic alternative to conventional antibiotics for the sterilization of explants and seeds. NPs can be incorporated directly into disinfection protocols or added to culture media to suppress microbial growth. Their antimicrobial action is based on multiple mechanisms, including disruption of microbial membranes, oxidative stress generation, and metal ion release, which limit the development of microbial resistance [124,138].

Silver NPs were among the first to be used for explant disinfection. In *Valeriana officinalis*, a combined treatment involving ethanol, sodium hypochlorite, and Ag NPs (100 mg/L) resulted in nearly complete decontamination without negatively affecting shoot proliferation or rooting [17]. Similarly, *Gerbera jamesonii* explants treated with NaOCl and 200 mg/L Ag NPs showed complete elimination of bacterial contamination and normal organogenesis [18]. In *Arabidopsis thaliana*, *Solanum tuberosum*, and *Lycopersicon esculentum*, surface disinfection using 100 mg/L Ag NPs proved sufficient for complete sterilization [139]. Furthermore, supplementation of Ag NPs or TiO_2_ NPs into the culture media significantly reduced contamination in potato and tobacco shoots, controlling endophytic infections in callus cultures. In *Bacopa monnieri*, the addition of 160 mg/L Ag NPs to the culture medium significantly reduced endogenous bacterial growth during subculturing [140]. More recently, hybrid Ag-ZnO nanocomposites have been explored as broad-spectrum disinfectants, offering synergistic antibacterial and antifungal properties at lower effective concentrations [141].

Disinfection of woody plant explants poses a particular challenge due to the presence of dense lignified tissues and abundant endophytic microorganisms. Traditional sterilization procedures often fail to achieve complete decontamination. For example, shoots from nine-year-old olive (*Olea europaea*) trees treated with ethanol and sodium hypochlorite showed only 48% survival free of contamination. However, the addition of Ag NPs to the protocol completely suppressed microbial growth, and the subsequent incorporation of Ag NPs into the culture medium prevented reinfection without any negative effect on morphogenesis [142]. Incorporating Ag, ZnO, or CuO NPs into disinfection protocols or tissue culture media not only improves sterility but can also enhance explant vigor, rooting, and regeneration efficiency by modulating oxidative and hormonal balance [143]. Nevertheless, NP concentration and exposure time must be carefully optimized, as excessive doses may induce phytotoxic effects, including growth inhibition, membrane damage, inhibition of germination, disruption of photosynthesis, and oxidative imbalance [144,145]. In some cases, NP exposure may also result in phytogenotoxic effects, which can occur either as a secondary consequence of pronounced phytotoxic stress or independently at subtoxic concentrations [146,147]. Genotoxicity refers to toxic effects on genetic material, manifested as DNA damage or chromosomal instability. Such effects may arise through direct interactions of NPs with genomic DNA or indirectly via oxidative damage mediated by reactive oxygen species (ROS) [148]. Future research should therefore focus on the development of standardized NP-based sterilization protocols tailored to specific plant species and explant types, enabling the safe integration of nanotechnology into sustainable micropropagation systems.

## 10. Effects of NPs on Callus Formation and Organogenesis

Due to their high reactivity and ability to penetrate cell walls and cytoplasmic membranes, metal NPs influence biochemical processes in plants [143,149]. As with most external agents, the impact, whether beneficial or detrimental, depends on the properties of the NPs, their concentration, and exposure duration (Figure 3). In *Tecomella undulata*, the efficiency of shoot regeneration and callus induction significantly increased when stem explants were cultivated on media supplemented with 10 mg/L Ag NPs [20]. The positive effect of Ag NPs on organogenesis may be attributed to their ability to inhibit ethylene production—a plant hormone that plays a crucial role in regulating plant growth and development. It has been shown that Ag NP treatment delays explant senescence and enhances survival by suppressing the expression of the ACS gene, which encodes acetyl-CoA synthetase [21]. Certain isoforms of this gene are involved in ethylene biosynthesis and cellular stress responses through interactions with the mitogen-activated protein kinase signaling pathway [21].

The addition of 50 mg/L Ag NPs to the culture medium significantly improved the growth parameters of *Brassica juncea* seedlings by reducing hydrogen peroxide and malondialdehyde levels through the activation of antioxidant enzymes [19]. However, higher concentrations of Ag NPs (100–400 mg/L) had detrimental effects on seedling growth. In *Brassica nigra*, the incorporation of ZnO NPs (500–1500 mg/L) into the medium significantly inhibited seed germination and affected shoot and root length [150]. Conversely, when *B. nigra* explants were cultured on media containing 1–20 mg/L ZnO NPs, root formation was not inhibited [150]. Kumar and colleagues [151] reported that adding Au NPs substantially increased seed germination rates and seedling growth in *Arabidopsis thaliana*. Furthermore, pod length and seed number were higher in plants treated with 10 mg/mL Au NPs.

The inclusion of Au NPs and Ag NPs in culture media with *Linum usitatissimum* stem segments enhanced embryogenesis by 70% and 50%, respectively [152]. The authors confirmed NP accumulation in plant cells and suggested that plants actively absorb metal NPs. However, the precise mechanism by which NPs enhance embryogenesis remains unclear and requires further investigation. Fazal and colleagues [153] examined the effects of individual gold and silver NPs, as well as their combinations, on callus proliferation in *Prunella vulgaris*. They found that Ag NPs (30 mg/L) and mixtures of Ag and Au NPs at 1:2 and 2:1 ratios, in combination with auxin, increased callus proliferation by 100% compared to controls.

## 11. Metal NPs as Elicitors

Plants have evolved complex defense networks that allow them to perceive environmental cues and respond by activating protective mechanisms and specialized metabolic pathways. Secondary metabolites such as phenolics, flavonoids, alkaloids, and terpenoids play central roles in plant adaptation, acting as antioxidants, antimicrobials, and signaling molecules (Figure 4). In plant biotechnology, in vitro systems, including callus, cell suspension, and hairy root cultures, are increasingly used as controllable platforms for the sustainable production of these valuable compounds. Recently, metal and metal oxide NPs have emerged as a new generation of abiotic elicitors that activate both defense and secondary metabolic responses in plants [15,154,155].

The biological effects of NPs are strongly influenced by their physicochemical properties, such as particle size, surface charge, and shape, which determine the mode of interaction with plant tissues, uptake efficiency, and intracellular localization [156]. Larger NPs are primarily retained at the cell wall or within the apoplastic space, where they can act as external elicitors by triggering membrane-associated signaling, redox imbalance, and stress-responsive pathways involved in secondary metabolism [157]. In contrast, smaller NPs can penetrate the cell wall pores and may be internalized into plant cells via intercellular transport, endocytosis-like processes, or plasmodesmata-mediated movement [158]. Once internalized, these NPs can directly interact with intracellular organelles, disturb redox homeostasis, and, at sufficiently small sizes, interact with nuclear components, potentially leading to DNA damage and genotoxic effects [148]. Surface charge plays a critical role in NP-plant interactions by influencing electrostatic interactions with the negatively charged components of the plant cell wall and plasma membrane. For example, positively charged NPs may be partially retained by the negatively charged cell wall matrix, which can affect their uptake and biological activity [159]. NP shape may further modulate cellular internalization and intracellular distribution, thereby influencing signaling intensity and downstream metabolic responses [159].

When plants are exposed to NPs, they undergo a transient oxidative burst, producing reactive oxygen species (ROS) and nitric oxide (NO) that act as secondary messengers in signal transduction [22,155]. These molecules activate calcium-dependent protein kinases (CDPKs) and mitogen-activated protein kinase (MAPK) cascades, key regulators of stress responses and secondary metabolism [160]. Activation of these signaling pathways leads to transcriptional upregulation of defense-related genes (e.g., WRKY, MYB, AP2/ERF) and biosynthetic enzymes such as phenylalanine ammonia-lyase (PAL), chalcone synthase (CHS), tyrosine decarboxylase, and 1-deoxy-D-xylulose-5-phosphate synthase (DXS), which drive the production of phenylpropanoids, flavonoids, alkaloids, and terpenoids [161].

Metal NPs also act as redox modulators and metabolic cofactors, directly interacting with enzymes to alter redox potential and NADPH availability. This enhances the activity of reductases and oxidases, stimulating secondary metabolism. For example, Fe-ZnO NPs increased total phenolic and flavonoid content and antioxidant capacity in *Fagonia indica* callus cultures [149]. Similarly, Ag NPs in *Catharanthus roseus* induced CrMPK3 and STR gene expression, boosting alkaloid accumulation via MAPK signaling [141].

Extensive experimental evidence confirms that NPs effectively enhance the biosynthesis of key secondary metabolites. Ag NPs promoted essential oil production in *Calendula officinalis* and increased phenolic accumulation in *Datura metel* hairy roots, outperforming classical elicitors such as AgNO_3_ or methyl jasmonate [151]. TiO_2_ NPs enhanced the biosynthesis of gallic, chlorogenic, and cinnamic acids in *Cicer arietinum* cultures, while bimetallic Au/Ag NPs increased flavonoid content in *Phaseolus vulgaris* calluses [44,162]. In *Hypericum perforatum* suspension cultures, diverse metal and metal oxide (Ag, CuO, ZnO, TiO_2_) NPs triggered the synthesis of over 100 novel secondary compounds, including bisxanthones, quercetin, and gallic acid [153,155]. NPs are also powerful activators of terpenoid pathways. In hairy root cultures of *Artemisia annua*, Ag NPs increased artemisinin yield up to fourfold [20], whereas Co NPs doubled production within 24 h by suppressing competing sterol biosynthetic genes (SQS, DBR2) [154].

NP exposure also primes the antioxidant and defense systems of plants. The ROS burst induces antioxidant enzymes such as superoxide dismutase (SOD), catalase (CAT), and peroxidases, and stimulates the biosynthesis of phytoalexins and pathogenesis-related (PR) proteins [143,149]. Fe_3_O_4_ NPs activated jasmonate-mediated defense signaling in *Withania somnifera* [112], increasing withanolide accumulation and oxidative stress tolerance, while CuO and ZnO NPs improved antioxidant capacity in *Mentha piperita* and *Ocimum basilicum* cultures [163,164]. In addition to their regulatory and elicitation effects, metal NPs play an important role in plant protection. For instance, biogenic ZnO, Ag, and CuO NPs exhibit strong antibacterial and antifungal activity against a wide range of phytopathogens, including *Pseudomonas syringae*, *Fusarium* spp., and *Alternaria* spp., thereby reducing disease incidence and improving plant resistance while minimizing the use of synthetic pesticides [165,166,167]. Such dual activation—defense and metabolism—reflects the integrated nature of NPs-mediated stress signaling.

## 12. Limitations and Future Prospects

The large-scale implementation of nanotechnology in agriculture and biotechnology holds significant potential for improving productivity, sustainability, and resource efficiency. In particular, when integrated into fertilization strategies, metallic NPs can act as advanced nutrient delivery systems, ensuring controlled release and higher bioavailability of essential elements [168]. However, despite rapid advances in nano-enabled fertilizers, pesticides, and genetic delivery systems, a growing body of evidence highlights substantial technical, ecological, and regulatory limitations that constrain their safe deployment at field and industrial scales. Understanding these constraints is essential for establishing sustainable nanotechnology-based practices in biological systems.

One of the foremost challenges concerns the ecotoxicological uncertainty of NPs under real environmental conditions. Laboratory studies frequently use simplified models that fail to represent the complex soil–plant–microbe interactions that govern NPs transport, transformation, and accumulation in agroecosystems. As a result, toxicity data remain fragmented and often contradictory. Silver, zinc oxide, and copper-based NPs have demonstrated both growth-promoting and inhibitory effects on plants and beneficial microorganisms, depending on concentration and environmental context [169,170]. In contrast to essential micronutrients such as Fe or Mn, silver does not participate in plant metabolic processes, and its biological effects are primarily associated with stress induction and antimicrobial activity. Consequently, the practical usefulness of Ag NPs is intrinsically concentration- and application-dependent, requiring a narrow balance between elicitation efficiency and toxicity. In particular, disruptions in mycorrhizal and rhizobial symbioses can weaken soil fertility and nitrogen fixation, threatening long-term ecosystem stability. Another critical limitation is the lack of standardized assessment protocols and regulatory frameworks. Current international guidelines for evaluating nano-agrochemicals are inconsistent, particularly regarding chronic exposure, NP persistence, and post-application behavior. Developing countries often lack infrastructure for nanowaste management, leading to uncontrolled releases of nanopollutants into soil and water systems [171]. Without harmonized risk-assessment standards, large-scale commercialization risks outpace safety validation.

Economic and scalability barriers further hinder the industrial production of biocompatible NPs. Traditional chemical synthesis routes rely on hazardous solvents and energy-intensive processes, limiting feasibility for small and medium-scale agricultural sectors. Although “green” synthesis methods using plant extracts or microbial systems have emerged, their reproducibility and yield remain inconsistent [172]. Moreover, NP formulation optimized under laboratory conditions often exhibits reduced stability and bioavailability when exposed to environmental stressors such as fluctuating pH, temperature, and salinity. Environmental sustainability presents another unresolved issue. Studies on nanoremediation highlight a dual role for NPs as agents for pollution cleanup and as potential contaminants themselves [173]. The high reactivity that makes NPs effective for contaminant degradation also increases their likelihood of interacting with non-target biota, raising concerns about bioaccumulation and trophic transfer. Additionally, most nanofertilizers and nanoformulations lack degradability, leading to potential buildup in agricultural soils after prolonged use.

To address these challenges, future research must emphasize multi-scale and multidisciplinary approaches. Long-term field studies that evaluate NP behavior under real agronomic conditions are urgently needed to complement laboratory-based toxicity assays. The development of biodegradable and “smart” NPs capable of self-deactivation after completing their function could mitigate accumulation risks. Likewise, integrating machine learning and environmental modeling could enhance predictive assessments of NP transport, transformation, and ecotoxicity [172]. Furthermore, harmonized international regulations should be established to govern NP production, application, and disposal. This includes creating databases linking physicochemical properties to ecological outcomes, enabling transparent risk communication among scientists, policymakers, and stakeholders. Finally, socioeconomic studies assessing farmer awareness, public perception, and cost–benefit ratios will be critical to bridge the gap between scientific innovation and field-level adoption.

## 13. Conclusions

The studies summarized in this review demonstrate that a wide spectrum of biological systems, including bacteria, fungi, algae, and cyanobacteria, whole plants, and in vitro plant cell cultures, can act as efficient biofactories for the green synthesis of metal and metal oxide NPs. Each platform provides distinct combinations of reducing and capping agents, from microbial enzymes and extracellular polymers to plant-derived polyphenols, proteins, polysaccharides, and secondary metabolites, enabling the production of NPs with diverse sizes, shapes, and compositions. However, these systems differ markedly in productivity, controllability, and biosafety. Microbial cultures are easy to handle but often show limited yields and potential pathogenicity. Algae and cyanobacteria are environmentally friendly but suffer from variable metabolite profiles. Whole plants offer high metal accumulation yet are strongly influenced by environmental conditions. In contrast, plant cell and tissue cultures emerge as particularly attractive platforms, combining high biosynthetic potential with strict control over growth conditions, metabolite composition, and NP characteristics. Recent advances highlight the importance of bioengineering the reductive capacity of biological systems as a new strategy to improve the efficiency and controllability of NP biosynthesis. Since the formation of NPs is primarily driven by key biomolecules—proteins, polysaccharides, and low-molecular-weight metabolites—targeted modification of their biosynthetic pathways can significantly influence NP yield and functional properties. This indicates that metabolic and genetic engineering of redox-active pathways can serve as a powerful tool for fine-tuning the biosynthetic potential of plants, enabling higher productivity and better control over the structural and functional properties of biogenic NPs.

Biogenic NPs produced by these systems have already found numerous applications in plant biotechnology. Metal and metal oxide NPs provide effective alternatives to conventional disinfectants and antibiotics for explant and seed sterilization, reducing microbial contamination while mitigating phytotoxicity and the risk of antibiotic resistance. When incorporated into culture media, they can modulate callus induction, organogenesis, and somatic embryogenesis through effects on ethylene signaling, redox balance, and antioxidant systems. Moreover, NPs function as potent abiotic elicitors that trigger reactive oxygen and nitrogen species signaling, activate MAPK and transcriptional networks, and ultimately enhance the accumulation of phenolics, flavonoids, alkaloids, terpenoids, and other high-value secondary metabolites in callus, suspension, and hairy root cultures. The size and shape of NPs play a decisive role in determining their interaction with plant cells and tissues. Smaller NPs (<50 nm) are generally more readily internalized through cell wall pores and plasmodesmata, leading to enhanced contact with intracellular organelles and greater modulation of redox and hormonal signaling pathways. Conversely, larger particles often remain adsorbed on the cell surface, acting primarily through indirect oxidative or osmotic effects. Particle morphology also influences biological responses: spherical NPs usually exhibit uniform uptake and moderate activity, whereas anisotropic forms possess higher surface energy and catalytic reactivity, often triggering stronger stress and defense responses. These physical attributes collectively determine NP-induced effects on callus induction, organogenesis, and overall morphogenetic potential in plant cultures. Beyond in vitro systems, biogenic NPs represent an emerging class of multifunctional agents for sustainable agriculture and biotechnology. Their use as nanofertilizers, nanopesticides, nanocarriers, and tools for phytoremediation provides new opportunities to enhance nutrient efficiency, stress tolerance, and environmental resilience. However, the translation of these technologies from controlled laboratory systems to real-world applications must be approached with caution and accompanied by rigorous evaluation of their ecological safety, degradability, and long-term behavior in agroecosystems.

## Figures and Tables

**Figure 1 plants-15-00183-f001:**
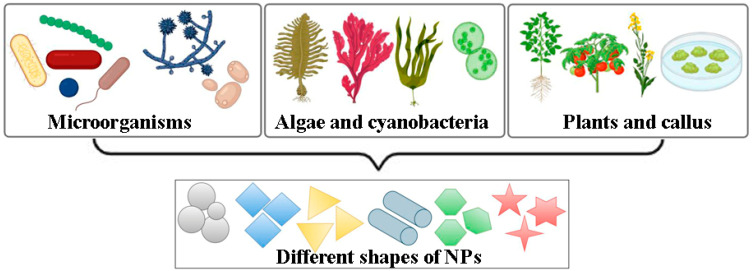
Biological systems used in the green synthesis of metal and metal oxide NPs. The figure illustrates various biological sources employed for biogenic NP production, including bacteria, fungi, algae, cyanobacteria, higher plants, and in vitro plant cell and tissue cultures. The particle shapes shown are schematic representations of common NP morphologies: spherical, cubic, triangular, hexagonal, and anisotropic shapes that can be encountered in biogenic systems.

**Figure 2 plants-15-00183-f002:**
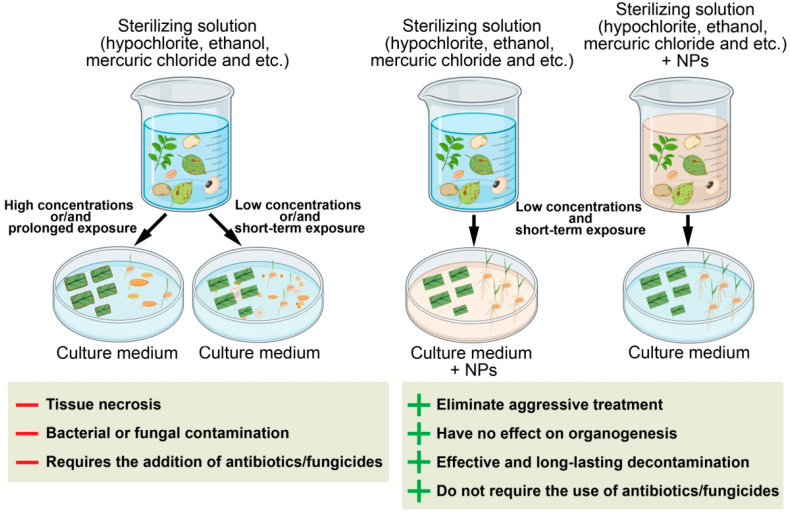
Schematic representation of explant and seed sterilization using metal NPs. The “+” and “–” symbols correspond to the advantages and disadvantages of the approaches, respectively.

**Figure 3 plants-15-00183-f003:**
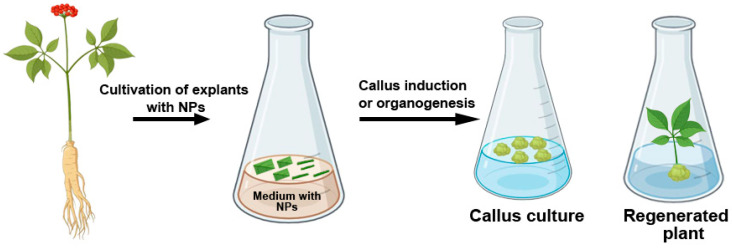
Effect of metal NPs on callus induction and organogenesis in plant tissue culture.

**Figure 4 plants-15-00183-f004:**
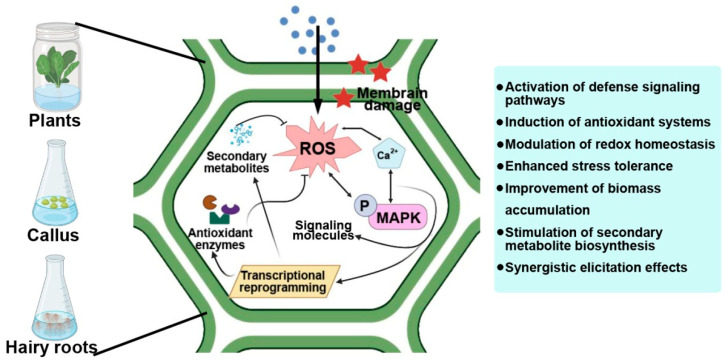
Mechanisms of secondary metabolism induction by NPs in plant cultures of calli and hairy roots.

**Table 1 plants-15-00183-t001:** Comparison between biogenic and physicochemical methods of metal NP synthesis.

Criteria	Biogenic Methods	Physico-Chemical Methods
Reducing and stabilizing agents	Natural biomolecules (enzymes, polysaccharides, proteins, flavonoids, etc.)	Synthetic reagents (NaBH_4_, hydrazine, citrate, etc.)
Reaction conditions	Mild temperature and pressure, aqueous media	High temperature/pressure, often organic solvents
Environmental impact	Eco-friendly, minimal waste, low toxicity	Generation of chemical waste, use of hazardous substances
Energy requirements	Low	High
Biocompatibility of NPs	High (suitable for biomedical and agricultural use)	Often requires post-synthesis surface modification
Size and shape control	Moderate, depends on biological system	High precision, but often less biocompatible
Scalability and reproducibility	Limited, dependent on biological variability	Highly scalable and reproducible
Example applications	Plant biotechnology, green catalysis, eco-friendly pesticides	Electronics, photonics, industrial catalysts

**Table 2 plants-15-00183-t002:** Obtaining metal and metal oxide NPs using bacteria.

Organism	Metal	Size (nm), Shape	References
*Pseudomonas rhodesiae*	Ag	20–100, spherical	[37]
*Bacillus siamensis*	Ag	25–50, spherical	[38]
*Bacillus cereus*	Ag	18–39, spherical	[39]
*Pseudomonas poae*	Ag	20–45, spherical	[40]
*Bacillus* sp.	Ag	7–21, spherical	[41]
*Serratia* sp.	Ag	10–20, spherical	[42]
*Stenotrophomonas* sp.	Ag	12, spherical	[43]
*Pseudomonas* sp. and *Achromobacter* sp.	Ag	20–50, spherical	[44]
*Streptomyces griseus*	Au	19–28, hexagonal	[45]
*Pseudomonas aeruginosa*	Au	7–39, spherical, triangular	[46]
*Salmonella enterica*	Au	42, spherical, crystalline	[47]
*Streptomyces* sp.	ZnO	20–50, hexagonal	[48]
*Paenibacillus polymyxa*	ZnO	56–110, cubic	[49]
*Aeromonas hydrophila*	ZnO	57–72, crystalline	[50]
*Streptomyces* spp.	CuO	78–80, spherical	[51]
*Streptomyces capillispiralis*	CuO	4–59, spherical	[52]
*Streptomyces pseudogriseolus*	CuO	78–80, spherical, crystalline	[51]
*Paenibacillus polymyxa*	MgO	10–19, spherical	[49]
*Paenibacillus polymyxa*	MnO_2_	20–64, spherical	[49]
*Gayadomonas* sp.	Te	15–23, rod-shaped nanostructures	[53]

**Table 6 plants-15-00183-t006:** Synthesis of metal and metal oxide NPs using plant cell cultures.

Organism	Metal	Size (nm), Shape	References
*Taxus yunnanensis*	Ag	6–27, spherical	[124]
*Hyptis suaveolens*	Ag	12–25, spherical	[125]
*Sesuvium portulacastrum*	Ag	5–20, spherical	[126]
*Linum usitatissimum*	Ag	19–54, spherical	[15]
*Solanum incanum*	Ag	15–60, spherical,	[127]
*Michelia champaca*	Ag, Au	5–9, spherical, triangular, oval	[128]
*Nicotiana tabacum*	Ag	20–80, spherical	[129]
*Lithospermum erythrorhizon*	Ag, Au, Ag/Au	10–45, spherical, triangular, oval	[16]
*Aristolochia manshuriensis*	Ag	10–40, spherical, oval	[130]
*Viola canescens*	ZnO	9–2, hexagonal	[123]

## Data Availability

No new data were created or analyzed in this study. Data sharing is not applicable to this article.

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
