# Peer review of "Biogenic Approaches to Metal Nanoparticle Synthesis and Their Application in Biotechnology"

_plants, 2026, doi:10.3390/plants15020183_

Round 1

Reviewer 1 Report

Comments and Suggestions for Authors

In this manuscript, the author provides a detailed introduction to the diverse biological sources and applications of metal nanoparticles. The article demonstrates clear logic, fluent language, and substantial content. The manuscript can be accepted by the journal after the following revisions are made.

  1. The author cited few references in the past five years. Please add more recent references.
    The author introduces that metal nanoparticles can be synthesized through fungal, bacterial and other pathways, and lists their biological sources, but does not provide a detailed description of the synthesis mechanism, which is not conducive to the in-depth understanding of readers. Further improvements are required.
    3. Metal nanoparticles have also been applied in plant protection, but this part is not included in the manuscript.

Author Response

In this manuscript, the author provides a detailed introduction to the diverse biological sources and applications of metal nanoparticles. The article demonstrates clear logic, fluent language, and substantial content. The manuscript can be accepted by the journal after the following revisions are made.

Response: We sincerely thank the reviewer for the positive evaluation of our manuscript and for recognizing its clarity, logical structure, and comprehensive content. We appreciate the constructive feedback and have carefully addressed all suggested revisions to further improve the quality and completeness of the paper.

Comment 1: The author cited few references in the past five years. Please add more recent references.

Response: We thank the reviewer for this important remark. We have updated the reference list by adding multiple recent publications from the past five years (2020–2025) to ensure that the manuscript reflects the most current research on biogenic synthesis and applications of metal nanoparticles.

Comment 2: The author introduces that metal nanoparticles can be synthesized through fungal, bacterial and other pathways, and lists their biological sources, but does not provide a detailed description of the synthesis mechanism, which is not conducive to the in-depth understanding of readers. Further improvements are required.

Response:  We thank the reviewer for this valuable comment. In the revised manuscript, we have substantially expanded the description of the biological mechanisms underlying metal nanoparticle synthesis. Specifically, a dedicated paragraph has been added to describe the general mechanistic framework of biologically mediated nanoparticle formation, including metal ion binding, bioreduction, nucleation, and particle growth, as well as the general roles of the main classes of biomolecules involved in this process. The specific enzymes and metabolites mentioned in this section are discussed in detail in subsequent chapters using representative biological systems as examples. This new information has been incorporated into the section entitled “Biological methods for metal NPs synthesis.”

Comment 3: Metal nanoparticles have also been applied in plant protection, but this part is not included in the manuscript.

Response: We thank the reviewer for this helpful comment. In response, we have added a short paragraph in the “Metal NPs as Elicitors” section discussing the role of metal nanoparticles in plant protection, including their antibacterial and antifungal activities against common phytopathogens such as Pseudomonas syringae, Fusarium spp. and Alternaria spp., supported by recent studies.

Reviewer 2 Report

Comments and Suggestions for Authors

The manuscript makes a favorable impression. The facts are presented quite clearly, without logical chains based on assumptions or illusions. The illustrations are high-quality. The review is easy to read and well-written.

  1. I would recommend supplementing the review with a table and a clear description of how biogenic methods for synthesizing nanoparticles differ from chemical/physical technologies. Experience shows that people are often misled.
  2. 2. I would recommend supplementing the review with a short chapter on nanofertilizers. I liked the review https://doi.org/10.3390/agronomy14081646, especially the comparison of the potency of classical fertilizers (K, Fe, Zn, and Mg) with nanosized fertilizers.

    Minor revision

Author Response

The manuscript makes a favorable impression. The facts are presented quite clearly, without logical chains based on assumptions or illusions. The illustrations are high-quality. The review is easy to read and well-written.

Response: We thank the reviewer for the kind and positive assessment of our work. We appreciate the constructive suggestions provided and have made the recommended revisions to enhance the manuscript.

Comment 1: I would recommend supplementing the review with a table and a clear description of how biogenic methods for synthesizing nanoparticles differ from chemical/physical technologies. Experience shows that people are often misled.
Response: We fully agree with the reviewer that a clear comparison between biogenic and chemical/physical nanoparticle synthesis methods would greatly benefit readers, especially those new to the field. To address this valuable suggestion, we have added a new comparative table (Table 1) and a short explanatory paragraph to highlight the fundamental distinctions between these approaches.

Comment 2: I would recommend supplementing the review with a short chapter on nanofertilizers. I liked the review https://doi.org/10.3390/agronomy14081646, especially the comparison of the potency of classical fertilizers (K, Fe, Zn, and Mg) with nanosized fertilizers.
Response: We thank the reviewer for this valuable suggestion and for highlighting the review by Semenova et al. (2024). Although our manuscript focuses on biogenic synthesis and plant biotechnological applications, we have added a concise sentence at the end of the “Limitations and Future Prospects” section to acknowledge the relevance of nanoscale nutrient formulations. This addition broadens the context by noting the potential of nanoparticles to enhance nutrient efficiency without altering the paper’s main scope.

Reviewer 3 Report

Comments and Suggestions for Authors

The review entitled “Biogenic Approaches to Metal Nanoparticle Synthesis and  Their Application in Biotechnology” is important for plant nanotechnology. However, the manuscript has omissions and requires major revisions.

-The author should describe the difference between phytotoxic and phytogenotoxic effects of NPs.

-In the sentence: “When introduced into culture media, NPs can enhance callus formation, organogenesis, and somatic embryogenesis by modulating ethylene biosynthesis, redox status, and antioxidant enzyme activity”, explain which specific nanoparticle you are referring to for these purposes; for example, AgNPs are for ethylene.

- A bibliometric analysis may be required

-Figure 1. Biological systems used in the green synthesis of metal and metal oxide nanoparticles. Why aren't all these shapes in the tables, especially the cylindrical shape? These have the appearance of nanotubes (nanomaterials) and not nanoparticles.

- Explain how the size and shape of NPs influence plant cells and tissues.

- In the subtitle: Application of NPs in Plant Biotechnology, the correct term is Application of NPs in Plant Tissue Culture

In Figure 4. Mechanisms of secondary metabolism induction by nanoparticles in plant cultures of calli and hairy roots., You only mentioned the nanoparticle size; explain how the size of the nanoparticle has different effects. You should discuss the interactions between plants and nanoparticles (NPs) (different types of absorption and transport mechanisms).  Example: Small nanoparticles can penetrate cells (intercellular movement). I recommend you read: Nanoparticle–Plant Interactions: Two-Way Traffic. For example, small nanoparticles can be internalized into cells and damage DNA.

A bibliometric analysis may be included.

Author Response

The review entitled “Biogenic Approaches to Metal Nanoparticle Synthesis and  Their Application in Biotechnology” is important for plant nanotechnology. However, the manuscript has omissions and requires major revisions.
Response: We thank the reviewer for the careful evaluation and recognition of the manuscript’s relevance to plant nanotechnology. We appreciate the constructive criticism and have thoroughly revised the paper to address all noted omissions and strengthen its overall quality.

Comment 1: The author should describe the difference between phytotoxic and phytogenotoxic effects of NPs. 
Response: We thank the reviewer for this comment. In the revised manuscript, we have clarified the distinction between phytotoxic and phytogenotoxic effects of nanoparticles, emphasizing their different biological levels and mechanisms of action. This information has been added to the section “Application of NPs in Plant Tissue Culture”.

Comment 2: In the sentence: “When introduced into culture media, NPs can enhance callus formation, organogenesis, and somatic embryogenesis by modulating ethylene biosynthesis, redox status, and antioxidant enzyme activity”, explain which specific nanoparticle you are referring to for these purposes; for example, AgNPs are for ethylene. 
Response: We thank the reviewer for this helpful comment. In the revised manuscript, we have clarified that the effects described in this sentence primarily refer to silver nanoparticles (Ag-NPs).

Comment 3: A bibliometric analysis may be required
Response: Corrected

Comment 3: Figure 1. Biological systems used in the green synthesis of metal and metal oxide nanoparticles. Why aren't all these shapes in the tables, especially the cylindrical shape? These have the appearance of nanotubes (nanomaterials) and not nanoparticles.

Response: We thank the reviewer for this insightful comment. In response, we have added several examples of nanoparticle morphologies and corresponding biological systems to the summary tables to better reflect the diversity of forms obtained through biogenic synthesis. Additionally, we have revised the caption of Figure 1 to clarify that cylindrical shapes represent anisotropic nanostructures rather than nanotubes and expanded the related paragraph to describe the variety of nanoparticle sizes and shapes produced in different biological systems.

Comment 4: Explain how the size and shape of NPs influence plant cells and tissues.
Response: We thank the reviewer for this valuable comment. We have added a concise explanation in the “Conclusion” section, describing how nanoparticle size and shape influence their cellular uptake, surface reactivity, and subsequent effects on plant growth and morphogenesis.

Comment 5: In the subtitle: Application of NPs in Plant Biotechnology, the correct term is Application of NPs in Plant Tissue Culture
Response: We thank the reviewer for the suggestion and have corrected the subtitle to “Application of NPs in Plant Tissue Culture.”

Comment 6: In Figure 4. Mechanisms of secondary metabolism induction by nanoparticles in plant cultures of calli and hairy roots., You only mentioned the nanoparticle size; explain how the size of the nanoparticle has different effects. You should discuss the interactions between plants and nanoparticles (NPs) (different types of absorption and transport mechanisms).  Example: Small nanoparticles can penetrate cells (intercellular movement). I recommend you read: Nanoparticle–Plant Interactions: Two-Way Traffic. For example, small nanoparticles can be internalized into cells and damage DNA. 
Response: We thank the reviewer for this constructive comment. In the revised manuscript, we have expanded the discussion to explain how nanoparticle size, surface charge, and shape influence plant–nanoparticle interactions, uptake pathways, and intracellular localization. The suggested reference, along with several recent related publications, has been incorporated into the revised version of the manuscript.

Comment 7: A bibliometric analysis may be included.
Response: We thank the reviewer for the suggestion and have added a brief bibliometric analysis to the revised manuscript.

Reviewer 4 Report

Comments and Suggestions for Authors

This review analyzes biological approaches to nanoparticle production using a wide range of biological systems. The reductive capacity of biological objects for the targeted synthesis of particles with desired characteristics is highlighted. The potential for biogenic nanoparticles to be used in various biotechnologies and agriculture is assessed. The authors believe that metal and metal oxide nanoparticles represent an effective alternative to traditional disinfectants and antibiotics. They hold promise for use in agriculture as nanofertilizers and nanopesticides. The authors draw upon a wide range of existing data in the relevant field of synthesis and application of nanoparticles obtained using "green chemistry." The conclusions they draw from this analysis are interesting, useful, and noteworthy. However, a number of the review's findings cannot be considered reliable, raise objections, and require further substantiation. Here are the main ones:

  1. The statement "Abstract: Biologically synthesized metal and metal oxide nanoparticles (NPs)..." is incorrect or, at the very least, questionable. NPs are produced by the chemical reduction of metal ions with components produced by biological objects (e.g., plant extracts), not by biochemical processes.
  2. The authors should have noted that the review, as well as the literature, synthesizes and discusses NPs of virtually only one metal—silver. All tables cited in the review list only silver. The data on Cu nanoparticles cited in the one table are likely incorrect. Copper metal particles are stable only in deaerated solutions. In the presence of air, they oxidize to form Cu2O, which is visually indistinguishable from the metal (yellow in color). The rare data on ZnO and Fe2O3 oxides are not related to redox synthesis. The authors should note that "Biogenic Approaches to Metal Nanoparticle Synthesis and Their Application in Biotechnology" essentially addresses the problem of synthesizing and using almost exclusively silver nanoparticles. This important question should be posed and answered.
  3. Many metals of “life” are involved in the biochemical processes of life – Fe, Co, Mn and many others. But not silver. This metal is like a two–faced Janus - a biocide on the one hand, and a strong toxin on the other. One is the effect and cause of the other. The question of usefulness is a question of concentration and method of application. This aspect of the issue should be within the scope of the authors' discussion.
  4. The widespread use of silver as a substitute for antibiotics will cause an increasing catastrophe of its spread and lead to its eternal and dangerous presence in the environment. Therefore, the authors, pointing to the broad prospects of application in agriculture, biotechnology and other fields, should discuss ways of subsequent utilization and recovery.

After taking into account the comments, the article can be recommended for publication in PLANTS.

Author Response

This review analyzes biological approaches to nanoparticle production using a wide range of biological systems. The reductive capacity of biological objects for the targeted synthesis of particles with desired characteristics is highlighted. The potential for biogenic nanoparticles to be used in various biotechnologies and agriculture is assessed. The authors believe that metal and metal oxide nanoparticles represent an effective alternative to traditional disinfectants and antibiotics. They hold promise for use in agriculture as nanofertilizers and nanopesticides. The authors draw upon a wide range of existing data in the relevant field of synthesis and application of nanoparticles obtained using "green chemistry." The conclusions they draw from this analysis are interesting, useful, and noteworthy. However, a number of the review's findings cannot be considered reliable, raise objections, and require further substantiation. Here are the main ones:
Response: We would like to sincerely thank you for your careful reading of our manuscript and for the valuable and constructive comments you have provided. We highly appreciate the time and effort you devoted to improving the quality of our work. We have carefully considered all your remarks and have provided detailed responses to each of them point by point in the revised version of the manuscript. Wherever necessary, we have made corresponding corrections and clarifications in the text to address your concerns and to strengthen the arguments and evidence presented in our review.

Comment 1: The statement "Abstract: Biologically synthesized metal and metal oxide nanoparticles (NPs)..." is incorrect or, at the very least, questionable. NPs are produced by the chemical reduction of metal ions with components produced by biological objects (e.g., plant extracts), not by biochemical processes.
Response: We thank the reviewer for this valuable comment and agree that clarification is required regarding the terminology describing the mechanisms of nanoparticle (NP) formation in biological systems. In the revised version, we have refined the wording in the Abstract to more accurately reflect the nature of the process. Although the reduction of metal ions during NP formation is indeed a chemical reaction, it is catalyzed or mediated by biological molecules (such as enzymes, proteins, polysaccharides, or phenolic compounds) that originate from living organisms or their extracts. Therefore, the term “biogenic” or “biologically mediated synthesis” is used in the literature to emphasize that these reactions occur under mild, biologically driven conditions rather than through conventional physicochemical methods.

Comment 2: The authors should have noted that the review, as well as the literature, synthesizes and discusses NPs of virtually only one metal—silver. All tables cited in the review list only silver. The data on Cu nanoparticles cited in the one table are likely incorrect. Copper metal particles are stable only in deaerated solutions. In the presence of air, they oxidize to form Cu2O, which is visually indistinguishable from the metal (yellow in color). The rare data on ZnO and Fe2O3 oxides are not related to redox synthesis. The authors should note that "Biogenic Approaches to Metal Nanoparticle Synthesis and Their Application in Biotechnology" essentially addresses the problem of synthesizing and using almost exclusively silver nanoparticles. This important question should be posed and answered.
Response: We thank the reviewer for this thoughtful and technically important comment. We agree that the literature shows a clear predominance of studies focused on silver nanoparticles, particularly in the section devoted to bacterial synthesis. This imbalance reflects a strong research bias toward AgNPs in biogenic nanoparticle synthesis and application studies. In the revised manuscript, we have revised and expanded the examples used throughout the review to better illustrate a broader range of elements whose nanoparticles can be produced using biological approaches, thereby providing a more balanced and representative overview of the field.
We also thank the reviewer for drawing attention to the incorrect use of copper nanoparticles in the original tables. We fully agree that metallic copper nanoparticles are unstable under aerobic conditions and readily oxidize. This issue has been carefully corrected in all relevant sections and tables of the revised manuscript.
We further agree that the formation of ZnO and Feâ‚‚O₃ nanoparticles in biological systems is generally not associated with classical redox-driven metal ion reduction. This clarification has been incorporated into the section “Biological methods for metal NPs synthesis” to avoid mechanistic ambiguity. However, in the present review, our primary focus is not on a detailed mechanistic classification of nanoparticle formation pathways, but rather on summarizing representative biogenic approaches and their applications in plant biotechnology.

Comment 3: Many metals of “life” are involved in the biochemical processes of life – Fe, Co, Mn and many others. But not silver. This metal is like a two–faced Janus - a biocide on the one hand, and a strong toxin on the other. One is the effect and cause of the other. The question of usefulness is a question of concentration and method of application. This aspect of the issue should be within the scope of the authors' discussion.
Response: We thank the reviewer for raising this important conceptual point. We agree that, unlike essential metals such as Fe, Co, or Mn, silver does not participate in physiological biochemical processes and its biological activity is primarily associated with stress induction and antimicrobial effects. From a biotechnological perspective, the relevance of AgNPs therefore lies not in their nutritional or metabolic role, but in their controlled use as abiotic elicitors or antimicrobial agents, where beneficial effects arise only within a narrow concentration window. In the revised manuscript, we emphasize that the usefulness of silver nanoparticles is intrinsically linked to dosage, exposure conditions, and application strategy, which determine whether AgNPs act as stimulatory signals or manifest toxic effects.

Comment 4: The widespread use of silver as a substitute for antibiotics will cause an increasing catastrophe of its spread and lead to its eternal and dangerous presence in the environment. Therefore, the authors, pointing to the broad prospects of application in agriculture, biotechnology and other fields, should discuss ways of subsequent utilization and recovery.
Response: We sincerely thank the reviewer for this important and constructive comment. We fully agree that the large-scale use of NPs raises significant environmental and biosafety concerns. In response, we have added a new section entitled “Limitations and Future Prospects”, where we now discuss in detail the ecotoxicological risks, regulatory limitations, and safe management strategies associated with the use of biogenic nanoparticles in agriculture and biotechnology.

Round 2

Reviewer 3 Report

Comments and Suggestions for Authors

Now, the authors have made significant changes to the manuscript. However, I cannot visualize the bibliometric analysis; this analysis is a figure created by some program (for example, VOSviewer). If you are unable to do so, explain the reason.

Author Response

Comment: Now, the authors have made significant changes to the manuscript. However, I cannot visualize the bibliometric analysis; this analysis is a figure created by some program (for example, VOSviewer). If you are unable to do so, explain the reason.

Response: We thank the reviewer for this comment. Unfortunately, due to limited access to major bibliographic databases such as Web of Science and Scopus in our country, we are currently unable to perform bibliometric visualization using specialized software such as VOSviewer. Therefore, we regret that we are unable to provide the requested bibliometric diagram in the revised version of the manuscript and ask for the reviewer's understanding.